# Plasma Arginase-1 Level Is Associated with the Mental Status of Outpatients with Chronic Liver Disease

**DOI:** 10.3390/diagnostics11020317

**Published:** 2021-02-16

**Authors:** Noriyoshi Ogino, Fusao Ikeda, Shihoko Namba, Shinnosuke Ohkubo, Tomoaki Nishimura, Hiroyuki Okada, Satoshi Hirohata, Narufumi Suganuma, Keiki Ogino

**Affiliations:** 1Department of Environmental Medicine, Kochi Medical School, Kohasu, Oko-cho, Nankoku City 783-8505, Japan; n-ogino@med.uoeh-u.ac.jp (N.O.); nsuganuma@kochi-u.ac.jp (N.S.); 2Third Department of Internal Medicine, School of Medicine, University of Occupational and Environmental Health, Iseigaoka 1-1, Yahatanishi-ku, Kitakyushu 807-8555, Japan; 3Department of Gastroenterology, Okayama University Graduate School of Medicine, Dentistry and Pharmaceutical Sciences, 2-5-1, Shikata-cho, Kita-ku, Okayama 700-8558, Japan; fikeda@md.okayama-u.ac.jp (F.I.); hiro@md.okayama-u.ac.jp (H.O.); 4Center for Innovative Clinical Medicine, Okayama University Hospital, Okayama, 2-5-1, Shikata-cho, Kita-ku, Okayama 700-8558, Japan; namba-s1@cc.okayama-u.ac.jp; 5Department of Medical Technology, Graduate School of Health Sciences, Okayama University, 2-5-1, Shikata-cho, Kita-ku, Okayama 700-8558, Japan; ohkubo-s@okayama-u.ac.jp (S.O.); hirohas@cc.okayama-u.ac.jp (S.H.); 6Micro Blood Science Inc., 2-14-8 Iwamotocho, Chiyoda-ku, Tokyo 101-0032, Japan; t.nishimura@microbs.jp

**Keywords:** mental status, arginase, liver disease

## Abstract

While plasma arginase-1 has been suggested as a biomarker of mental status in healthy individuals, it has not been evaluated in patients with chronic liver disease. This cross-sectional study investigated the utility of plasma arginase-1 for screening mental status in patients with chronic liver disease. This study included outpatients with chronic liver disease who underwent regular check-ups at Okayama University Hospital between September 2018 and January 2019. In addition to the standard blood tests, the plasma arginase-1 level was analyzed. The patients’ mental status was assessed using the Japanese version of the General Health Questionnaire-28 (GHQ-28). The associations between mental status and various parameters, including plasma arginase-1, were investigated using logistic regression analysis. Among 114 participating patients, 8 were excluded, comprising 6 with insufficient blood samples for plasma arginase-1 measurement and 2 with incomplete questionnaires. Multivariate binomial logistic regression analysis revealed that plasma arginase-1 was significantly and negatively associated with the GHQ-total score, especially somatic symptoms. Therefore, plasma arginase-1 may be a useful biomarker for assessing the mental status of outpatients with chronic liver disease.

## 1. Introduction

Stress plays a role in the development of physical and mental disorders [1,2,3]. A physical response to stress occurs owing to an imbalance in the autonomic nervous system; moreover, the associated chronic inflammation contributes to the development of physical diseases [4,5,6,7]. However, questionnaires are exclusively used to measure stress, and few “gold standard” measures of stress can be used simply in daily practice [1]. Stress is experienced in multiple ways, including social, psychological, and physiological stress, making it challenging to establish biomarkers for some aspects of stress.

Chronic liver diseases (e.g., viral hepatitis, non-alcoholic fatty liver disease, and alcoholic liver disease) and cirrhosis, the end-stage of all chronic liver diseases, interfere with mental and physical well-being, as well as daily activities [8,9,10,11]. Moreover, a meta-analysis of the association between psychological distress and liver disease mortality showed a significant increase in liver disease mortality with increased General Health Questionnaire (GHQ) score [12]. Therefore, assessing the mental and physical status of patients with chronic liver disease in daily practice is an important factor for the treatment of liver disease.

We previously found that arginase-1, interleukin-6, and C-reactive protein might be useful as stress indicators and candidate biomarkers of chronic inflammation [13,14,15,16,17]. Arginase, a key enzyme in the urea cycle, is involved in the indirect regulation of nitric oxide (NO) by the consumption of L-arginine, which is a common substrate for NO synthase (NOS) [18]. NO from NOS in neuronal cells acts as a neurotransmitter and modulates norepinephrine, serotonin, dopamine, and glutamate; thus, disruption of NO metabolism leads to psychiatric disorders [19]. Similarly, plasma arginase levels and activity are also associated with psychiatric disorders, as arginase regulates NO metabolism [20]. Serum arginase-1 (one of two iso-enzymes) levels were significantly associated with oxidative stress, exhaled NO, and L-arginine in a healthy population [13,14,15] and were a significant explanatory variable for job strain in healthy workers [21]. Although these results suggest that blood arginase-1 levels could be an indicator of mental status in healthy individuals, arginase-1 has not been explored in patients with chronic liver disease. Therefore, the present cross-sectional study investigated the relationship between arginase-1 levels and the mental status of patients with chronic liver disease.

## 2. Materials and Methods

### 2.1. Study Design

This cross-sectional study included outpatients with chronic liver disease at the hepatitis clinic of Okayama University Hospital between September 2018 and January 2019. We finally analyzed 106 patients who completed the Japanese version of the General Health Questionnaire-28 (GHQ-28) and in whom we could measure plasma arginase-1 levels to determine the relationship between mental status and clinical parameters. The study was conducted following the principles of the Declaration of Helsinki and was approved by the ethical committees of Okayama University Graduate School of Medicine, Dentistry and Pharmaceutical Sciences and Okayama University Hospital (#1611–04). All patients provided written informed consent.

### 2.2. Sample Collection

Medical staff collected whole blood and plasma or serum samples simultaneously from the participants in fasting state in the morning. All samples were stored at 4 °C during their transfer to the clinical laboratory.

### 2.3. Biochemical and Blood Tests

The following parameters were evaluated using JCA-BM8040, JCA-BM6070 (JOEL Ltd., Akishima, Japan), and ADVIA 2120 (Hematology System, Siemens Healthcare Diagnostics) instruments: white blood cell, neutrophil, lymphocyte, red blood cell, and platelet counts; prothrombin time; and albumin, aspartate transaminase (AST), alanine transaminase (ALT), γ-glutamyl transferase (GTP), lactate dehydrogenase (LDH), triglyceride, total cholesterol, low-density lipoprotein cholesterol (LDL-C), high-density lipoprotein cholesterol (HDL-C), uric acid, blood urea nitrogen, creatinine, and hemoglobin A1c (HbA1c) concentrations. Plasma arginase-1 was measured using an enzyme-linked immunosorbent assay for human liver-type arginase (BioVendor, Heidelberg, Germany) according to the manufacturer’s protocol and was re-measured by gradual dilution if the values were saturated.

### 2.4. GHQ-28 Assessment of Stress Response in Outpatients with Chronic Liver Disease

We used the Japanese version of the GHQ-28, a self-administered screening questionnaire designed for use in primary care settings [22]. The questionnaire explores four dimensions—somatic symptoms (GHQ-A), anxiety and insomnia (GHQ-B), social dysfunction (GHQ-C), and depression (GHQ-D)—with a list of 28 items, each one rated on a 4-point Likert-type scale: “not at all”, “no more than usual”, “rather more than usual”, and “much more than usual”. We used two scoring methods: “not at all” = 0, “no more than usual” = 0, “rather more than usual” = 1, and “much more than usual” = 1.

### 2.5. Assessment of Liver Status

In order to compare the liver status of patients with various etiologies, we analyzed the pathological data obtained from recent liver biopsy specimens (*n* = 93) assessed based on the New Inuyama Classification [23]. The stage of fibrosis (F) was defined as follows: F0 (no fibrosis), F1 (fibrosis evident as portal expansion), F2 (bridging fibrosis), F3 (bridging fibrosis with lobular distortion), or F4 (cirrhosis). Disease activity (A) was defined as follows: A0 (no necro-inflammatory reaction), A1 (mild necro-inflammatory reaction), A2 (moderate necro-inflammatory reaction), and A3 (severe necro-inflammatory reaction). We also analyzed the data of liver stiffness measurements (*n* = 52) obtained from transient elastography via FibroScan^®^ within one year [24].

### 2.6. Statistical Analysis

Associations with each scale score and the outpatient variables were examined using Spearman’s rank correlation coefficients to identify factors related to mental status evaluated in the GHQ-28. One-way ANOVA was used for the comparison of each liver etiology for GHQ score and arginase-1. Binominal logistic regression analysis was conducted using major background factors and the parameters that had significant associations (*p* < 0.05) in the analysis with the GHQ-28 score as the dependent variables. Statistical analysis was performed using Graph Pad Prism 5 (Graph Pad Software Inc., San Diego, CA, USA) and PASW Statistics 18.0 (SPSS Inc., Chicago, IL, USA).

## 3. Results

Among 114 patients who participated, 8 were excluded (6 with insufficient blood samples for plasma arginase-1 measurement and 2 with incomplete questionnaires). Thus, 106 outpatients were analyzed. Their etiologies of liver disease, liver status, and clinical characteristics are shown in Table 1 and Table 2 and in Appendix A. Regarding patient characteristics, 42% were male, and the mean (± SD) age was 61.1 (±13.2) years. Among all patients, 35 (33%) had the hepatitis B virus and 37 (35%) had the hepatitis C virus (one patient was co-infected). Four patients presented histologically confirmed cirrhosis (F4). Although the mean of the liver stiffness measurements of 52 patients was 6.379 kPa, the measurements of 5 patients were above 12.5 kpa, which is the standard value for the diagnosis of cirrhosis [24]. Twelve patients had diabetes mellitus, 33 had hypertension, and 29 had dyslipidemia in addition to chronic liver disease. Fifty participants followed an exercise routine, 32 were drinkers, and 25 were smokers. The mean total GHQ-28 score was 5.08 ± 4.43 using the two scoring methods. The mean values of blood parameters, including liver inflammation and fibrosis, such as ALT, platelet counts, and prothrombin time, were all within the normal ranges.

A significant inverse correlation was observed between the GHQ-total score and arginase-1 or creatinine levels in the Spearman’s correlation coefficient analysis (Table 3). For each item in the GHQ, a significant positive correlation was observed between GHQ-D and white blood cells (WBCs), whereas significant negative correlations were observed between GHQ-A and arginase-1 or creatinine and between GHQ-C and age. Moreover, significant correlations were observed between arginase-1 and sex, neutrophil-to-lymphocyte ratio, and creatinine level, in addition to GHQ-total and GHQ-A. There was no significant correlation observed between liver status and arginase-1 or GHQ score. No differences were observed in patients among hepatitis B and C viruses (HBV, HCV) and others (non-HBV and non-HCV) for arginase-1 and GHQ score (Appendix A). Although the patients’ comorbidities, smoking habit, and alcohol drinking did not have a significant correlation with any GHQ score, exercise habit was significantly negatively correlated with GHQ-A and GHQ-C scores (Appendix A). The results of a multivariate binomial logistic regression analysis to evaluate the association between the GHQ-total score or GHQ-A score and arginase-1 are shown in Table 4. In the sex- and age-adjusted analysis, patients with higher plasma arginase-1 levels tended to have a lower stress response (*P*-trend < 0.004); the odds ratio (OR) (95% confidence interval (CI)) was 0.129 (0.034–0.486) when comparing the top and bottom quartiles (Q4 vs. Q1). After further adjusting for body mass index (BMI), white blood cell count, neutrophil-to-lymphocyte ratio (NLR), red blood cell count, platelet count, albumin concentration, ALT concentration, creatinine concentration, and exercise habit, the OR remained significant (*P*-trend = 0.005) but was attenuated to 0.081. Similarly, a significant inverse association was observed between GHQ-A and arginase-1, in which a higher plasma concentration of arginase-1 was associated with significantly lower GHQ-A score (*P*-trend < 0.012). The adjusted OR (95% CI) for Q4 vs. Q1 was 0.030 (0.020–0.820) in multivariate-adjusted Model 3.

## 4. Discussion

To our knowledge, this is the first report to assess plasma arginase-1 for the screening of mental status in outpatients with chronic liver disease.

In this study, the lower the arginase-1 level, the more severe the somatic symptoms. Several studies have reported an association between chronic liver disease and somatic symptoms [8,9,10,11,12]. One of the most common and incapacitating symptoms experienced, especially in patients with hepatitis C, is depression-induced fatigue and somatic complaints [25,26]. Although no study has directly described the involvement of arginase-1 with physical symptoms, it is similar to the negative relationship previously reported between workload and arginase-1 in healthy adults [21]. Plasma arginase-1 accounts for most of the plasma arginase activity and regulates the metabolism of L-arginine and NO [18]. Mental stress increases the levels of adrenocortical hormones and decreases arginase-1 expression. Although more detailed studies are needed, there may be an association between physical symptoms and NO metabolism [27].

It is known that cirrhosis impairs mental status due to hepatic encephalopathy [28]. In this study, the patients were not examined for hepatic encephalopathy. However, considering that all outpatients were examined by two hepatologists, wrote a statement of understanding and consent for the study, and answered the GHQ questionnaire, they at least did not have overt hepatic encephalopathy. Hence, the patients could be considered as either subclinical hepatic encephalopathic or normal. Since subclinical hepatic encephalopathy might affect the GHQ score, a detailed evaluation of subclinical hepatic encephalopathy is desirable. In addition, arginase is an enzyme related to ammonia metabolism [18], and its relationship with subclinical hepatic encephalopathy and GHQ score needs to be analyzed in more detail, which is a subject for further study [29,30]. Regarding the diagnosis of cirrhosis in this study, eight patients were diagnosed as having cirrhosis based on stage 4 liver fibrosis or a liver stiffness measurement of >12.5 kPa [24]. Three more patients could be added to this list if we consider suspected cirrhosis, determined via the following criteria: prothrombin time (PT)% less than 70% or albumin concentration less than 3.5 g/dL, total bilirubin concentration more than 2 mg/dL (more than 4 mg/dL for primary biliary cholangitis), or platelet count less than 100,000/μL [31]. Since we did not confirm the presence of hepatic encephalopathy and ascites, this was not a correct assessment, but 11 patients were suspected to have cirrhosis in this study. A statistical comparison of the arginase-1 and GHQ scores of the two groups (11 suspected cirrhosis patient group and other group) showed no significant difference. These results were not consistent with those of previous studies [12,32]. This might be due to insufficiency of the sample size of cirrhotic patients in this study.

Assessing the physical and mental health of patients with chronic liver disease is as important as treating the disease [12,33]. In addition to controlling the disease state, various factors are known to affect the quality of life of patients with chronic liver disease [34]. The presence of chronic liver disease itself was reported to be the cause of psychological problems [35]. Although the mechanisms of how mental stress affects physical symptoms are still unclear, this relationship is considered to be another important factor [36]. Low levels of chronic inflammation, as indicated by C-reactive protein (CRP) or Interleukin-6 (IL-6), have been reported as a factor explaining this relationship, but the mechanism is not fully understood [16,17]. Therefore, questionnaires such as the GHQ, Short Form 36 (SF-36), and the Chronic Liver Disease Questionnaire (CLDQ) have been used to assess the mental status of patients, but their use in daily practice is cumbersome [37,38]. Studies have assessed cortisol and blood ammonia levels in the saliva and hair of patients with chronic liver disease to identify a simple method for assessing patient mental status that can be used in daily practice; however, these assessments have not been applied clinically [39,40]. Plasma arginase-1 may be more useful for measuring the mental status of patients with chronic liver disease because it is relatively stable during the day and can be measured with routine blood tests.

The limitations of this study included the small sample size and the fact that it was a single-center study. Although previous studies reported an association between smoking habit or alcohol drinking and GHQ score [41], no significant correlation was observed between any GHQ score and these lifestyle factors or comorbidities in this study. We also did not find differences in any GHQ score with regard to etiology of liver disease. However, when we examined the correlations of GHQ split scores with exercise habit, GHQ-A and GHQ-C were significantly correlated. These results were consistent with another report [42]. If the sample size had been larger, similar results may have been obtained for smoking habit or alcohol drinking. Second, this study did not include a control group. Patients with chronic liver disease reportedly have compromised mental health compared with healthy individuals. When assessing GHQ score, it is difficult to make simple comparisons with other healthy individuals because there is so much variability among the target populations [43,44]. Third, there are limited research data on human plasma arginase-1 levels; thus, there are no standards. Although it was not a formal comparison, the plasma arginase-1 levels in this study tended to be slightly higher than those of healthy individuals of the same age group reported previously [13]. While plasma arginase-1 was reported to be an indicator of liver damage [45], no significant association was observed between levels of plasma arginase-1 and liver enzymes such as ALT in the present study. These differences may be related to the fact that the liver enzymes among the subjects in our study were within the normal ranges, except for γ-GTP, which exceeded the upper limit of normal. Therefore, plasma arginase-1 may be of extrahepatic origin [46,47].

Despite these limitations, it is important to identify indicators for monitoring mental status in patients with chronic liver disease [48,49]. Plasma arginase-1 may be useful for screening the mental status of outpatients with chronic liver disease.

## Figures and Tables

**Table 1 diagnostics-11-00317-t001:** The etiologies and fibrosis stages of liver disease among the patients.

Characteristic	Number (%)	
Total ^a^	106	
Male	44	(41.5)
Hepatitis B virus	35	(33.0)
Hepatitis C virus	37	(34.9)
Autoimmune hepatitis	8	(7.5)
Primary biliary cholangitis	11	(10.4)
NAFLD ^b^	7	(6.6)
Others	10	(9.4)
Liver biopsy (Fibrosis stage ^c^)	93	
F0	69 (74.1)	
F1	12 (12.9)	
F2	3 (3.2)	
F3	5 (5.4)	
F4	4 (4.3)	

^a^ One patient presented co-infection of hepatitis B and C viruses, and another presented an overlap of primary biliary cholangitis and autoimmune hepatitis. ^b^ NAFLD, non-alcoholic fatty liver disease. ^c^ Fibrosis stages are based on the New Inuyama Classification. Degree of fibrosis (F): F0 (no fibrosis), F1 (fibrosis evident as portal expansion), F2 (bridging fibrosis), F3 (bridging fibrosis with lobular distortion), F4 (cirrhosis).

**Table 2 diagnostics-11-00317-t002:** Patient clinical measurements.

Variable	Mean ± SD		*n*
GHQ-A score (somatic symptoms)	2.01 ± 1.83		106
GHQ-B score (anxiety/insomnia)	2.15 ± 1.8		106
GHQ-C score (social dysfunction)	0.53 ± 1.1		106
GHQ-D score (depression)	0.38 ± 1.21		106
GHQ-Total score	5.08 ± 4.43		106
**Variable**	**Unit**	**Mean ± SD**	*****n*****
Age	years	61.1 ± 13.2	105
Height	cm	161 ± 9.04	98
Bodyweight	kg	60.35 ± 12.5	99
BMI		23.28 ± 4.01	98
White blood cell	/μL	4800 ± 1560	105
Neutrophil	%	56.9 ± 10.40	93
Lymphocyte	%	33.1 ± 9.63	93
Red blood cell	×10^6^/μL	4.4 ± 0.50	105
Platelet	×10^4^/μL	21 ± 8.81	105
Albumin	g/dL	4.21 ± 0.37	106
Prothrombin time	%	101 ± 16.5	64
AST	U/L	26.9 ± 12.7	106
ALT	U/L	22.7 ± 20.2	106
GTP	U/L	51.1 ± 95.8	105
LDH	U/L	195 ± 36.8	105
Triglycerides	mg/dL	104 ± 61.2	88
Total cholesterol	mg/dL	202 ± 37.3	100
HDL-c	mg/dL	66 ± 19.2	78
LCL-c	mg/dL	120 ± 30	85
Uric acid	mg/dL	5.21 ± 1.49	101
Blood urea nitrogen	mg/dL	15.5 ± 6.97	106
Creatinine	mg/dL	0.93 ± 1.78	101
HbA1c	%	5.5 ± 0.61	85
Arginase-1	ng/mL	27.48 ± 52.78	106
Liver stiffness	kPa	6.379 ± 4.63	52

GHQ, General Health Questionnaire; BMI, body mass index; AST, aspartate transaminase; ALT, alanine transaminase; GTP, γ-glutamyl transferase; LDH, lactate dehydrogenase; HDL, high-density lipoprotein cholesterol; LDL, low-density lipoprotein cholesterol; HbA1c, hemoglobin A1c. Liver stiffness was examined by transient elastography via FibroScan^®^.

**Table 3 diagnostics-11-00317-t003:** Spearman’s correlation of each GHQ score and arginase-1 with several clinical parameters.

Variable	GHQ-A		GHQ-B		GHQ-C		GHQ-D		GHQ-Total		Arginase-1	
	r	P	r	P	r	P	r	P	r	P	r	P
Sex (M/F)	0.131	0.18	0.084	0.390	0.105	0.286	−0.126	0.198	0.096	0.326	−0.219	**0.026**
Age	0.127	0.137	−0.174	0.075	−0.305	**0.002**	−0.141	0.153	−0.177	0.072	0.121	0.218
BMI	0.02	0.843	−0.12	0.24	0.022	0.827	0.171	0.092	−0.035	0.730	0.008	0.934
White blood cell	0.066	0.503	0.063	0.526	−0.021	0.835	0.202	**0.039**	0.096	0.330	0.036	0.713
NLR	0.094	0.369	−0.027	0.977	0.025	0.815	0.086	0.411	0.057	0.590	0.282	**0.0062**
Red blood cell	−0.068	0.492	−0.069	0.487	−0.028	0.774	0.101	0.307	−0.059	0.550	0.044	0.653
Platelet	0.124	0.209	0.137	0.164	0.102	0.299	0.124	0.206	0.193	**0.049**	−0.110	0.27
Albumin	0.028	0.777	0.123	0.209	−0.156	0.110	0.063	0.520	0.113	0.251	−0.008	0.932
ALT	−0.006	0.802	0.071	0.472	0.123	0.208	0.165	0.090	0.081	0.408	0.051	0.605
Prothrombin time	−0.055	0.668	0.018	0.888	−0.229	0.069	−0.056	0.658	0.016	0.897	0.184	0.145
Creatinine	−0.292	**0.003**	−0.152	0.121	−0.181	0.064	−0.006	0.951	−0.256	**0.008**	0.247	**0.011**
Arginase 1	−0.339	**<0.001**	−0.090	0.361	−0.171	0.079	−0.014	0.884	−0.235	**0.015**	-	-
Fibrosis stage	−0.033	0.757	0.036	0.729	0.052	0.621	−0.024	0.816	0.056	0.591	0.163	0.119
Activity stage	−0.032	0.761	0.024	0.816	0.01	0.924	−0.056	0.596	0.027	0.797	0.118	0.261
Liver stiffness	0.140	0.322	0.218	0.120	0.202	0.151	0.256	0.067	0.213	0.130	0.081	0.571

GHQ, General Health Questionnaire; BMI, body mass index; NLR, neutrophil-to-lymphocyte ratio; ALT, alanine transaminase; GHQ-A, score of somatic symptoms; GHQ-B, score of anxiety/insomnia; GHQ-C, score of social dysfunction; GHQ-D score of depression; Significant correlations (*p* < 0.05) are denoted in bold. Liver stiffness was examined by transient elastography via FibroScan^®^. Fibrosis and activity stage are based on the New Inuyama Classification.

**Table 4 diagnostics-11-00317-t004:** Odds for GHQ-Total and GHQ-A according to arginase-1 concentration.

Explanatory Variable	Quartiles of Arginase-1 (Odds for GHQ-Total)				P Trend
	Q1	Q2	Q3	Q4	
Model 1	1	0.504 (0.168–1.511)	0.546 (0.184–1.619)	**0.140 (0.040–0.489)**	**0.003**
Model 2	1	0.451 (0.142–1.438)	0.573 (0.190–1.731)	**0.129 (0.034–0.486)**	**0.004**
Model 3	1	0.391 (0.085–1.709)	0.327 (0.082–1.310)	**0.081 (0.014–0.471)**	**0.005**
Model 1: Odds for GHQ-Total according to arginase-1 with no adjustment.					
Model 2: Odds for GHQ-Total according to arginase-1 adjusted for age and sex.					
Model 3 (*N* = 85): Odds for GHQ-Total according to arginase-1 adjusted for age, sex, BMI, WBCs, NLR, RBCs, platelets, albumin, ALT, creatinine, and exercise habit.					
Explanatory variable	Quartiles of arginase-1 (Odds for GHQ-A)				P trend
	Q1	Q2	Q3	Q4	
Model 1	1	0.790 (0.264–2.335)	0.308 (0.095–1.002)	**0.140 (0.034–0.581)**	**0.002**
Model 2	1	0.883 (0.277–2.820)	**0.294 (0.087–0.994)**	**0.169 (0.038–0.744)**	**0.006**
Model 3	1	0.642(0.136–3.040)	0.241 (0.056–1.038)	**0.030 (0.020–0.820)**	**0.012**
Model 1: Odds for GHQ-A according to arginase-1 with no adjustment.					
Model 2: Odds for GHQ-A according to arginase-1 adjusted for age and sex.					
Model 3 (*N* = 85): Odds for GHQ-A according to arginase-1 adjusted for age, sex, BMI, WBCs, NLR, RBCs, platelets, albumin, ALT, creatinine, and exercise habit.					

GHQ, General Health Questionnaire; BMI, body mass index; WBCs, white blood cells; NLR, neutrophil-to-lymphocyte ratio; RBCs, red blood cells; ALT, alanine transaminase. Q1, Q2, Q3, Q4: Quartiles of log_10_ format of arginase-1 with ranges of ≤0.775610400 (Q1), 0.775610401–1.126996 (Q2), 1.169996001–1.389166 (Q3), and ≥1.389166001 (Q4). Significant correlations (*p* < 0.05) are denoted in bold.

## Data Availability

The data presented in this study are available on request from the corresponding author. The data are not publicly available due to the possibility of identifying certain members of the population.

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
