# Peer review of "Plasma Arginase-1 Level Is Associated with the Mental Status of Outpatients with Chronic Liver Disease"

_diagnostics, 2021, doi:10.3390/diagnostics11020317_

Round 1

Reviewer 1 Report

The present study of Ogino et al. investigated the relationship of mental health and plasma Arginase 1 in patients with chronic liver disease.  In brief, this study found the negative correlation between mental health status measured with a questionnaire and plasma  Arginase 1.  Unfortunately, the authors do not give any information about the status of the liver disease.  Especially, it is not known if there are patients with cirrhosis included and if there are patients with cirrhosis how many patients at  what stage of cirrhosis are included.  It is known that cirrhosis of the liver impairs the mental status due to hepatic encephalopathy.  There are established tests to check for even subclinical encephalopathy and it would be of interest if These tests actually correlate with the plasma levels of Arginase 1 and the health questionnaire.

Related questions would be:  Is there a correlation to the fibrosis;  is there a correlation to disease  activity, duration of disease ,treatments of the disease? 

In order to make this study  valuable an analysis   of cirrhotic patients and encephalopathy should be included  and discussed. and encephalopathy

Reviewer 2 Report

The paper; “Plasma arginase-1 level is associated with the mental status of outpatients with chronic liver disease” by Noriyoshi Ogino et al. measured plasma arginase-1 in plasma of 106 patients with chronic liver diseases and found correlations with measures of mental health.

„chronic viral hepatitis, fatty liver, and cirrhosis” may be viral hepatitis, non-alcoholic fatty liver disease and alcoholic liver disease. Fatty liver is also a feature of HCV infection and cirrhosis is the end-stage of all chronic liver diseases.

“l-arginine“ please correct.

“used to measure stress and few "gold standard" measures of stress“ please list and explain these gold standards.

Page 2, “General Health Questionnaire“ please explain, it is obvious that people having worse disease have a higher score.

“We previously found that arginase-1, as well as interleukin-6 and C-reactive protein,

might be useful as stress indicators and candidate biomarkers of chronic inflammation” Association of IL-6 and CRP with chronic inflammation was shown by various groups, please clarify.

At what day time was plasma / serum collected. Was this done in the fasted state?

“Regarding patient characteristics, 42% were female” and in table 1 42% were males.

Table 2, authors may give values of GHQ scores for healthy individuals.

Significant associations may be shown in bold or colour in table 3.

Please explain Q1 to Q4 also in the table legends.

Table 4 “–1 (Odds for GHQ-total)” please correct.

Is arginase-1 induced in patients with chronic liver disease in comparison to healthy controls?

“However, we observed no difference in GHQ scores by comorbidity, lifestyle, or etiology of liver disease in the present study.” These data should be included.

Round 2

Reviewer 1 Report

Within the scope of the study, encephalopathy has been discussed.The limitations of the study have been described.

Author Response

Thank you very much for your kindly suggestion. We will do further research on plasma arginase and subclinical hepatic encephalopathy.

Reviewer 2 Report

Still, there are no references for CRP and IL-6 as inflammatory markers. 

Regarding Response 4 authors say: "However, there is no mechanistic consensus to explain the dexterous effects of psychological problems on physical health." It is likely that being ill may cause psychological problems.  This should be discussed in the paper. 

Lymphocyto, stifffness, … please correct. 

Are there any threshold values for the scores used?
